# The Development of a Human Respiratory Mucosa-on-a-Chip Using Human Turbinate-Derived Mesenchymal Stem Cells

**DOI:** 10.3390/medicina60111741

**Published:** 2024-10-24

**Authors:** Do Hyun Kim, Sang Hi Park, Mi-Yeon Kwon, Chae-Yoon Lim, Sun Hwa Park, David W. Jang, Se Hwan Hwang, Sung Won Kim

**Affiliations:** 1Department of Otolaryngology-Head and Neck Surgery, College of Medicine, The Catholic University of Korea, Seoul 06591, Republic of Korea; 2Institute of Clinical Medicine Research, College of Medicine, Catholic University of Korea, Seoul 06591, Republic of Korea; 3Postech-Catholic Biomedical Engineering Institute, College of Medicine, The Catholic University of Korea, Seoul 06591, Republic of Korea; 4Department of Head and Neck Surgery & Communication Sciences, Duke University School of Medicine, Durham, NC 27710, USA

**Keywords:** microfluidics, respiratory system, stem cells, culture, humans

## Abstract

*Background and Objectives*: This study aimed to investigate the influence of a respiratory mucosa-on-a-chip on the respiratory epithelial differentiation potential of human nasal inferior turbinate-derived stem cells (hNTSCs). *Materials and Methods*: After isolating hNTSCs from five patients, we divided the samples from the patients into the study group with a mucosa-on-a-chip and the control group with conventional differentiation (using conventional differentiation methods). The respiratory epithelial differentiation potential of hNTSCs was analyzed by histology and gene expression. *Results*: In the quantitative analysis, PCR showed that the hNTSCs expressed the cytokeratin genes (KRT13, 14), transformation-related protein P63 (TP63), and vimentin of basal cells in the airway epithelium at higher levels, but cytokeratin genes (KRT6) at lower levels, in the mucosa-on-a-chip than in conventional differentiation. In the cytokine analysis (GM-CSF, IFNr, IL-1a, IL-1b, IL-4, IL-5, IL-10, IL-12(p70), IL-13, IL-17A, IL-17E/IL-25, RANTES, TNFa, IL-6, and IL-8), the expressions of IFNr, IL-13, RANTES, TNFa, IL-6, and IL-8 were significantly upregulated in the mucosa-on-a-chip than in conventional differentiation. *Conclusions*: We conclude that the human respiratory mucosa-on-a-chip using human turbinate-derived mesenchymal stem cells allows the respiratory differentiation of hNTSCs and shows the difference in gene and cytokine expression, which could serve as an alternative to conventional differentiation for the production of functionally competent hNTSCs for future clinical applications.

## 1. Introduction

The airway mucosa is the primary site of contact for inhaled materials, such as allergens, microorganisms, viruses, and air pollutants, thereby providing protection to the airways [1]. Additionally, it acts as a barrier against harmful environmental agents [2]. Studying the airway epithelium is essential for understanding the underlying mechanisms of various respiratory diseases and conditions. Damage to the nasal epithelial and submucosal structures is implicated in the development of nasal disorders, including polyps, sinusitis, and cystic fibrosis [3,4,5]. Moreover, mucociliary clearance serves as an intrinsic defense mechanism that helps the airways fend off external irritants. Cell differentiation can be induced or promoted by cellular mechanical cues, spatial arrangements, and biomaterials [6]. Therefore, several microfluidic airway-on-a-chip techniques have been developed to create air–liquid interface (ALI) culture conditions [7,8,9]. However, these reports mainly evaluated the functional similarity to humans and did not discuss differentiation methods that were more effective than conventional culture methods. We hypothesized that utilizing on-chip respiratory mucosa could create a more suitable environment for cell differentiation, thereby creating functionally superior airway mucosa.

Mesenchymal stem cells (MSCs) are a type of multipotent stem cell that can be derived from a variety of tissues. Their unique characteristics, including self-renewal, the ability to differentiate into multiple cell lineages, and immunomodulatory capabilities, make them a valuable cellular resource for regenerative medicine. Beyond their potential to directly differentiate into tissue-specific cells at sites of injury, MSCs exhibit significant paracrine activity by secreting trophic factors and extracellular vesicles. These secretions play a crucial role in modulating the microenvironment and facilitating tissue repair and regeneration [10]. However, the in vitro expansion of MSCs often faces challenges, such as cell death, cellular senescence, and a reduction in multipotency [11]. As a result, research has focused on identifying optimal culture conditions to improve the quality of MSCs, including media composition, oxygen tension, or substrate coatings [12,13,14]. Human nasal inferior turbinate-derived stem cells (hNTSCs) can be readily harvested during the treatment of chronic rhinitis and have been identified as a valuable source of MSCs [15,16,17]. hNTSCs demonstrate strong resilience against environmental conditions, exhibit high proliferative capacity, and consistently retain their characteristics across multiple cell passages [18,19]. Despite the recognition of hNTSCs as a promising source of MSCs, the specific effects of microfluidic airway-on-a-chip conditions on hNTSCs have not yet been investigated.

Therefore, in the current study, we introduced an innovative human upper respiratory mucosa-on-a-chip model to investigate the influence of the respiratory mucosa-on-a-chip on the respiratory epithelial differentiation potential of human nasal turbinate-derived hNTSCs.

## 2. Materials and Methods

This study was conducted in compliance with the guidelines established by the Institutional Review Board of the Catholic Medical Center Clinical Research Coordinating Center (HC15TISI0022), adhering to informed consent regulations and the principles set forth in the Declaration of Helsinki. Written informed consent was obtained from all participants prior to surgery, and all procedures received approval from the Institutional Review Board. Informed consent was directly acquired from each participant.

### 2.1. Donors and Cell Isolation

Inferior turbinate tissues were collected from 5 patients after the age of 20 undergoing partial turbinectomy and assigned to the mucosa-on-a-chip differentiation (study group) and control (using conventional differentiation methods) groups. Patients with nasal polyposis and congenital immunologic diseases were excluded.

hNTSCs were isolated from the collected inferior turbinate tissues. The tissue samples were initially rinsed thoroughly with an antibiotic–antimycotic solution (Gibco, Gaithersburg, MD, USA) three to five times, followed by two washes with phosphate-buffered saline (PBS). Subsequently, the tissues were sectioned into small fragments of approximately 1 mm^3^. These tissue pieces were then placed in culture dishes coated with CELLstart CTS Attachment Substrate (Gibco), in accordance with the manufacturer’s instructions, and covered with a sterilized glass coverslip. StemPro^®^ MSC SFM XenoFree medium (Gibco), supplemented with 200 mM of L-glutamine (Gibco), was added to the culture dishes, which were incubated at 37 °C in a 5% carbon dioxide environment. The culture medium was changed every 2–3 days. After a 3-week incubation period, the glass coverslips were removed, and any floating tissue was washed away. The remaining human nasal stem cells (hNTSCs), which had adhered to the bottom of the culture dish, were detached using TrypLE Select 10× (Gibco). The hNTSCs from four passages were subsequently analyzed for culture media-related changes in immunophenotypic characteristics and respiratory differentiation.

### 2.2. Characterization by Analysis of Cell Surface Markers on hNTSCs

Flow cytometry analysis was conducted to assess the cell surface markers on hNTSCs. The cells were prepared at a density of 1 × 10⁵ cells/mL and placed in test tubes (BD, Franklin Lakes, NJ, USA), followed by three washes with phosphate-buffered saline (PBS). The cells were then incubated for 1 h with primary antibodies, including monoclonal antibodies CD14, CD29, CD34, CD73, CD90, and HLA-DR, all obtained from BD Biosciences (San Jose, CA, USA), at saturating concentrations. After incubation, the cells underwent three additional washes with the buffer and were centrifuged at 400× *g* for 5 min. The cells were then resuspended in ice-cold PBS and incubated with secondary antibodies in the dark at 4 °C for 30 min. Flow cytometry was performed using a FACSCaliber Flow Cytometer (BD), and the resulting data were analyzed using the CellQuest software v5.1 (BD).

### 2.3. Respiratory Differentiation Potential of hNTSCs on Respiratory Mucosa-on-a-Chip

The human respiratory mucosa-on-a-chip model was engineered in-house, comprising a mucociliary epithelium and a stromal component, specifically designed to establish an air–liquid interface (ALI). The ALI culture device was constructed from three main components: an upper chamber, a lower chamber, and an electrospun nanofibrous membrane. The electrospun nanofibrous membrane was positioned between the upper and lower chambers. The membrane was fabricated in the following manner: polycaprolactone was dissolved in a 9:1 chloroform–dimethylformamide solution and subsequently electrospun into nanofibers with diameters ranging from 300 to 500 nm, resulting in the formation of a nanofibrous membrane with a thickness of 200 µm. The electrospinning process was conducted at room temperature, utilizing a flow rate of 2 mL/h and an applied voltage of 18 kV. The inner diameter of the needle was 0.4 mm, and the distance between the needle tip and the collecting plate was maintained at 15 cm. Before seeding, the mucosa-on-a-chip was treated with Eo gas, coated with plasma for 1 to 2 min, and exposed to UV light for 5 min, and the hNTSCs were cultured in a growth medium (DMEM low-glucose, 10% FBS, 2 mM GlutaMAX, and 5 μg/mL gentamicin (10 mg/mL stock)) for 1 h. The hNTSCs were seeded on the membrane (2 × 10^5^ cell/well), and the cells were grown submerged in seeding media (DMEM low-glucose, 10% FBS, 2 mM GlutaMAX, 5 μg/mL gentamicin, 30 ng/mL EGF, and 10 ng/mL KGF/FGF-7) in a humidified 5% CO_2_ atmosphere at 37 °C for 2 days. These chips were then differentiated under ALI conditions with differentiation media (DMEM low-glucose, 2% FBS, 2 mM GlutaMAX, 5 μg/mL gentamicin, 10 ng/mL HGF, 60 ng/mL IGF, 30 ng/mL EGF, and 10 ng/mL KGF/FGF-7) for 21 days. After 21 days, the cultures were monitored for differentiation by using lineage-specific biological stains. In the control group, for comparison, hNTSCs were seeded on 4-well chamber slides (1.7 cm^2^/well), grown, and differentiated under the same conditions without using the mucosa-on-a-chip.

To monitor respiratory differentiation, the cells underwent histological examination and immunofluorescent staining and were observed under an inverted microscope. In addition, the gene expression according to the differentiation was analyzed by RT-PCR.

### 2.4. Cytokines Assays

hNTSCs were seeded at a density of 2 × 10⁵ cells per well in 12-well plates (24 plates in total) and allowed to adhere overnight. Conditioned media collected from both microchip and control conditions on day 21 were stored at −80 °C. These media samples were then analyzed using the MILLIPLEX MAP human cytokine/chemokine multiplex immunoassay (Millipore, Billerica, MA, USA) to quantify various interleukins (IL-1α, IL-1β, IL-4, IL-5, IL-6, IL-8, IL-10, IL-12, IL-13, IL-17A, and IL-17E/IL-25), RANTES (CCL5), tumor necrosis factor (TNF)-α, granulocyte-macrophage colony-stimulating factor (GM-CSF), and interferon (IFN)-γ, in accordance with the manufacturer’s instructions. These analyses were performed independently on at least three separate occasions using different MSC donor pools.

### 2.5. RNA Extraction and RT-PCR of hNTSCs

Total RNA was isolated from the epithelial cells grown in the mucosa-on-a-chip using the QIAzol lysis reagent (QIAGEN, Valencia, CA, USA). In brief, the membrane placed between the upper and lower body chip device was rinsed with cold PBS, and the membrane separated from the two chip devices was collected in a tube and homogenized with 300 µL of QIAzol lysis regent using a MagNaLyser (Roche, Mannheim, Germany). The homogenized cells were collected, and phenol/chloroform extraction was carried out following the manufacturer’s instructions. The isolated RNA was quantified with a Biophotometer D30 (Eppendorf, Hamburg, Germany). Subsequently, 625 ng of purified RNA was reverse-transcribed into first-strand complementary DNA using the CellScript cDNA Synthesis Master Mix (CellSafe, Suwon, Republic of Korea), which incorporates a genomic DNA elimination step. The cells were cultured under respiratory differentiation conditions in a conventional 2D culture system and lysed with 350 μL of QIAzol lysis reagent, with the resulting lysates analyzed using the same protocol. Real-time polymerase chain reaction (RT-PCR) was utilized for the amplification and relative quantification of cytokeratin genes (KRT6, KRT13, KRT14, and KRT18), transformation-related protein P63 (TP63), and vimentin employing TaqMan gene expression assays (Table 1) (Applied Biosystems, Foster City, CA, USA) with a LightCycler 480 PCR system (Roche, Mannheim, Germany). The assays demonstrated consistent amplification efficiency, and a delta cycle threshold method was applied for relative quantification. The reactions were conducted in triplicate in a 10 μL volume using a TaqMan Probe Master Mix (Roche), with 125 ng of complementary DNA per reaction. Glyceraldehyde 3-phosphate dehydrogenase (GAPDH) was used as the endogenous control. Data analysis was conducted using the LightCycler 480 Instrument Software 1.2 (Roche).

### 2.6. Histological Examination and Immunofluorescent Staining

The collected and fixed samples were subjected to a dehydration process through a graded ethanol series, followed by clearing with xylene and embedding in paraffin. Paraffin sections, 5 µm in thickness, were cut from the central region of each sample, mounted on glass slides, allowed to dry overnight, and stored at 4 °C. These sections were subsequently stained with hematoxylin and eosin (H&E) using an H&E Staining Kit (Abcam, Cambridge, MA, USA) for general morphological assessment.

For the immunofluorescent staining of the hNTSCs, the sections were deparaffinized with xylene and rehydrated through a graded ethanol series. Antigen retrieval was achieved using a proteinase K solution (Abcam) to uncover antibody-binding sites. The sections were then incubated with primary antibodies against MUC5AC (ab218466, mouse, 1:50), MUC2 (ab134119, rabbit, 1:50), cytokeratin 18 (ab32118, rabbit, 1:50), and cytokeratin 19 (ab7754, mouse, 1:50). Subsequently, the sections were incubated with secondary antibodies, including Alexa Fluor 546 (green)- or Alexa Fluor 488 (red)-conjugated goat anti-rabbit or anti-mouse IgG (1:1000; Molecular Probes, Eugene, OR, USA).

### 2.7. Statistical Analysis

Statistical analyses were conducted using the R statistical software v4.4.1 (R Foundation for Statistical Computing, Vienna, Austria). To evaluate the statistical significance of the differences between groups, *t*-tests and one-way analysis of variance (ANOVA) were applied. We conducted a study with cells donated by 5 participants, and the sample size for all results was 5. A *p*-value of less than 0.05 was considered to indicate statistical significance.

## 3. Results

### 3.1. Characterization of hNTSCs

In both groups, the hNTSCs were negative for hematopoietic markers (CD14, CD34, and HLA-DR) and positive for mesenchymal stem cell markers (CD29, CD73, and CD90) (see Figure 1 and Figure 2), thereby reflecting the typical phenotype associated with mesenchymal stem cells.

### 3.2. Cytokine and Chemokine Secretion Patterns of hNTSCs in Mucosa-on-a-Chip and Conventional Culture Group by Flow Cytometry Compared with Those in Control Group

We measured a number of cytokines and chemokines known to be involved in immunomodulation, including IL-1α, IL-1β, IL-4, IL-6, IL-8, IL-10, IL-12, IL-13, IL-17A, IL-17E/IL-25, RANTES, TNF-α, GM-CSF, and IFN-γ. Among them, IFNr, IL-1a, IL-1b, IL-10, IL-12(p70), IL-13, IL-17E/IL-25, RANTES, TNFa, IL-6, and IL-8 were detectable with mean values of more than 1 pg/mL in supernatants harvested from hNTSCs in both groups. However, some cytokines and chemokines showed significant differences in the level of secretion in the mucosa-on-a-chip group compared with the control group. The mucosa-on-a-chip strongly induced the expressions of IFNr, IL-13, RANTES, TNFa, IL-6, and IL-8. In contrast, the secretion of IL-17E/IL-25 was significantly reduced in the mucosa-on-a-chip group compared with the control group. Specifically, an approximately 10- to 40-fold difference in the expressions of IFNr, RANTES, IL-6, and IL-8 was observed with the serum-free culture. IL-13 and TNFa expression showed a difference of two- to five-fold, respectively (Figure 3). Overall, these results suggest that hNTSCs are immunologically influenced according to the culture methods.

### 3.3. Respiratory Differentiation Potential of hNTSCs in Mucosa-on-a-Chip and Conventional Culture Groups

The in-house-developed human respiratory mucosa-on-a-chip model was composed of a mucociliary epithelium and a stromal component, specifically engineered to maintain an air–liquid interface (Figure 4 and Figure 5). In the first preliminary test to verify the adequate environment for hNTSCs, hNTSCs were cultivated in the mucosa-on-a-chip in growth media for 2 weeks, and these cells proliferated well with multi-layered growth and attachment to the membrane of the chip (Figure 6A). In another preliminary test to verify the effectiveness of the differentiation conditions, hNTSCs were cultivated under the respiratory differentiation conditions of a conventional 2D culture for 3 weeks. These cells proliferated well with the immunostaining of cytokeratin 18, cytokeratin 19, and mucin 5AC. (Figure 6B). In the respiratory differentiation of the mucosa-on-a-chip, hNTSCs were cultivated for 3 weeks, and these cells proliferated well with multi-layered growth and attachment to the membrane of the chip (Figure 6C). In the immunofluorescence stain, hNTSCs were stained with cytokeratin 18, cytokeratin 19, MUC2, and mucin 5AC (Figure 6D). The specific respiratory differentiation potential of the expanded hNTSCs in the two groups was determined by histologic approaches to detect the staining of cytokeratin 18, cytokeratin 19, MUC2, and mucin 5AC (Figure 6E). The respiratory differentiation conditions induced differentiation-specific markers in the monolayers of cells from the two groups, as revealed by immunofluorescence staining. Additionally, quantitative analysis of the gene expression of factors regarding differentiation was determined by RT-PCR. Cells exposed to the differentiation medium expressed mRNAs encoding cytokeratin genes (KRT6, 13, 14, and 18), transformation-related protein P63 (TP63), and vimentin of basal cells in the airway epithelium. However, there were significant differences in the differentiation capacity of hNTSCs derived from both conditions. Cytokeratin genes (KRT 13 and 14), transformation-related protein P63 (TP63), and vimentin of basal cells in the airway epithelium in the mucosa-on-a-chip group were significantly upregulated compared with those in the control group. In contrast, cytokeratin genes (KRT6) in the mucosa-on-a-chip group were significantly downregulated compared with those in the control group.

## 4. Discussion

MSCs have garnered considerable research attention because of their therapeutic potential in regenerative medicine and cell therapy. These cells are characterized by their multipotency, enabling them to differentiate into a range of cell types, including osteocytes, chondrocytes, and adipocytes. Additionally, their immunomodulatory properties and ability to secrete bioactive molecules make them suitable for treating a range of inflammatory and degenerative diseases. Despite these advantages, translating in vitro findings into effective in vivo therapies has proven challenging, as preclinical studies often report inconsistencies between laboratory and clinical outcomes.

One significant challenge in the application of MSCs is their expansion and differentiation in vitro. During in vitro culture, MSCs frequently encounter issues such as reduced proliferation, cellular senescence, and loss of multipotency, which can affect their therapeutic efficacy [20]. Various factors influence MSC culture, including the composition of the growth medium, oxygen tension, and the extracellular matrix [21,22]. Optimizing these conditions is crucial to maintaining the quality and functionality of MSCs for therapeutic purposes.

hNTSCs represent a valuable source of MSCs, particularly because they can be easily harvested during routine surgical procedures for chronic rhinitis. These cells have demonstrated high resilience to environmental stress and a strong proliferation potential, maintaining their characteristics across multiple cell passages. However, while hNTSCs have been recognized for their MSC-like properties, the influence of different culture conditions, including the use of advanced models like microfluidic airway-on-a-chip systems, on their differentiation and functional capacity has not been thoroughly investigated.

The novel respiratory mucosa-on-a-chip model introduced in this study offers a promising platform for investigating the differentiation potential and functional responses of hNTSCs under more physiologically relevant conditions. This model aims to create an environment that closely resembles the in vivo respiratory mucosa. This approach is expected to provide insights into the behavior of hNTSCs in a more native-like context, potentially overcoming some of the limitations associated with traditional 2D culture systems. The results of fluorescence staining may appear similar between the 2D culture and chip model, but the chip model was able to produce more diverse gene expression trends. In the results of this analysis, the 2D culture showed higher results in staining, and the 3D culture showed higher results in gene expression. This shows that the form of differentiation proceeds differently depending on the differentiation environment between the two methods, but this does not mean that one method is superior. It would be necessary to elucidate these differences in protein expression and genetic expression through additional studies in the future.

The upper respiratory mucosa consists of the superficial epithelium, submucosal glands, subepithelial stromal cells, and endothelial cells, bound together by apical junctional complexes, including adherens and tight junctions [23]. The epithelium plays a crucial role in maintaining cell integrity and regulating the movement of ions, water, and molecules to ensure homeostasis [24]. Damage to the airway epithelial barrier compromises cell junction integrity and increases the vulnerability of the submucosa to allergens, toxins, and microorganisms [2,24]. To replicate the upper respiratory system in artificial models, it is necessary to incorporate these factors [20]; however, recreating functional environments that closely resemble in vivo conditions remains a significant challenge.

In this study, the respiratory mucosa-on-a-chip model demonstrated enhanced cytokine activity and a more robust differentiation response compared with conventional culture methods. The upregulation of specific cytokines and chemokines, such as IL-6, IL-8, and TNF-α, suggests that the model successfully recapitulates some of the key immune functions of the respiratory mucosa. These findings are significant as they indicate that the model can be used to study the interactions between respiratory epithelial cells and various external stimuli, including pathogens and environmental toxins. The expression of markers like cytokeratin 18, cytokeratin 19, and mucin 5AC further confirmed the successful differentiation of hNTSCs into respiratory epithelial cells. This is particularly important for applications in regenerative medicine, where the ability to generate a functional respiratory epithelium is essential for the treatment of diseases such as cystic fibrosis, chronic obstructive pulmonary disease, and asthma.

The translation of the respiratory mucosa-on-a-chip model into clinical applications presents several significant challenges. First, biological complexity is a primary concern. While the chip can replicate certain physiological features of the respiratory mucosa, the human respiratory system involves a highly dynamic interplay of immune responses, microbial interactions, and variable environmental exposures. Replicating this complexity on a chip, particularly in a way that accounts for the heterogeneity of patient populations, remains difficult. Another challenge is maintaining the long-term viability and functionality of the cellular components on the chip, as in vitro models often deteriorate over time, impacting their reliability for prolonged testing and therapeutic evaluation.

Another potential barrier is regulatory approval and validation. Before clinical use, any in vitro model must undergo rigorous testing to demonstrate its predictive accuracy in human conditions. This requires extensive comparative studies between the chip model and human trials, which can be time-consuming and costly. Furthermore, there are ethical and safety concerns regarding the use of these models to simulate human diseases, especially when considering immunocompromised or highly vulnerable patient groups.

Regarding the scalability and practical use of the respiratory mucosa-on-a-chip, there are technical and logistical challenges to be addressed. Manufacturing the chips at the scale necessary for widespread clinical application involves overcoming obstacles related to standardization and quality control. Each chip should consistently replicate the same biological conditions to ensure reproducibility across experiments and applications. Moreover, cost is a significant factor. Producing highly sophisticated and reliable models at an affordable price point for widespread clinical or pharmaceutical use is not trivial, particularly as the models become more complex to simulate specific patient conditions.

Lastly, the integration of this model into current clinical workflows poses practical concerns. Healthcare facilities would need to be equipped with the technical expertise and equipment to utilize these chips effectively, which may not be feasible in all clinical settings. Training personnel and integrating the chip model into routine diagnostic or therapeutic procedures will require time and resources, further complicating its transition to clinical use.

Using a novel human respiratory mucosa-on-a-chip model, we were able to generate a model with higher cytokine activity functionally than conventional differentiation methods. This model is anticipated to be a valuable tool for assessing the respiratory mucosa’s responses to external bacteria, viruses, and inhaled medications. However, this model also has limitations. First, this chip was intended to mimic respiratory mucosa, but submucosa tissue was not implemented. It was constrained by the absence of other physiological components, such as immune cells, that are present in native tissues. In the actual human submucosa, small vessels exist, and through these, immune cells and inflammatory cells interact with each other. Additionally, glands such as Bowman’s glands exist in the submucosa and secrete mucus, but these are absent in this model simulating the epithelium. However, there are still limitations in implementing such real human responses in microfluidics. Several studies have attempted to induce interactions by co-culture with immune cells or by creating microvascular structures in the substructure, but there are still limitations in producing responses similar to those in the human body. Second, this initial experiment was limited by an insufficient sample size. Despite these limitations, the model is anticipated to be valuable for evaluating the respiratory mucosa’s response to external bacteria, viruses, and inhaled drugs.

## 5. Conclusions

In conclusion, the novel human respiratory mucosa-on-a-chip model developed in this study represents a significant advancement in the field of respiratory research. By providing a more physiologically relevant environment for the differentiation and functional assessment of hNTSCs, this model has the potential to enhance our understanding of respiratory diseases and to facilitate the development of new therapies. However, further research is needed to fully realize its potential and to address the current limitations, such as biological complexity, regulatory approval, scalability, and practical integration. As the field of tissue engineering and regenerative medicine continues to evolve, models like the respiratory mucosa-on-a-chip will play an increasingly important role in bridging the gap between in vitro studies and clinical applications.

## Figures and Tables

**Figure 1 medicina-60-01741-f001:**
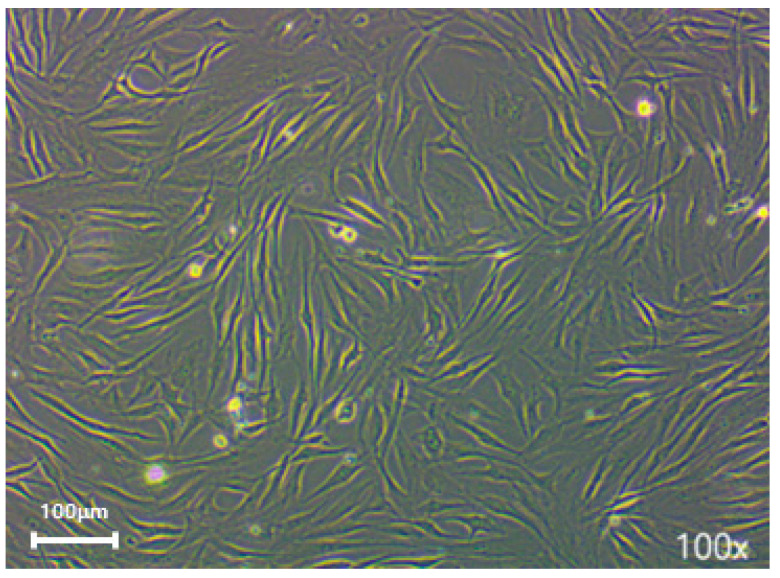
Illustrates the morphology following primary explant culture, The cells in both groups adhered to the culture dish and displayed a spindle-shaped, fibroblast-like morphology (magnification: ×100).

**Figure 2 medicina-60-01741-f002:**
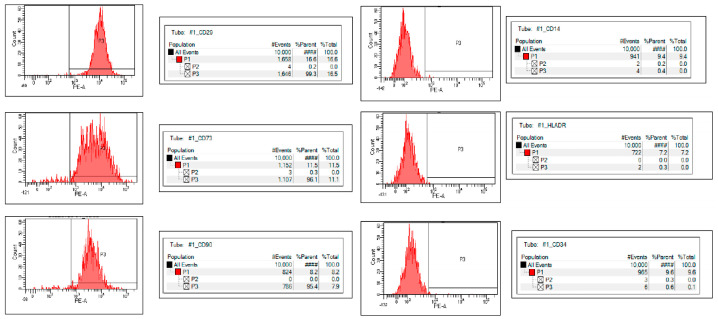
Flow cytometry analysis, performed after three passages, verified that human nasal stem cells (hNTSCs) from both groups were positive for CD29, CD73, and CD90 and negative for CD14, CD34, and HLA-DR. #### means 100%.

**Figure 3 medicina-60-01741-f003:**
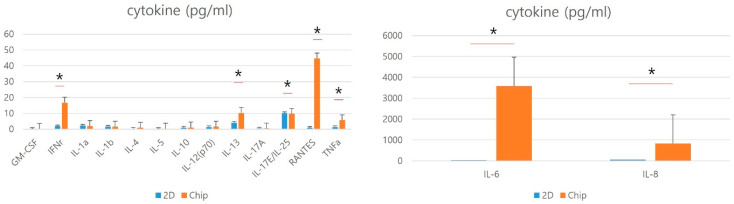
Effects of serum-free cultivation on cytokine and chemokine secretion by hNTSCs. Secretion of cytokines and chemokines such as IL-1α, IL-1β, IL-4, IL-6, IL-8, IL-10, IL-12, IL-13, IL-17A, IL-17E/IL-25, RANTES, TNF-α, GM-CSF, and IFN-γ of hNTSCs from mucosa-on-a-chip and conventional culture were evaluated with enzyme-linked immunosorbent assay. In hNTSCs derived from mucosa-on-a-chip, there were upregulated expressions of IFNr, IL-13, RANTES, TNFa, IL-6, and IL-8 and downregulated expression of IL-17E/IL-25 compared with control group. These patterns in the secretion of cytokines and chemokines were different from those of hNTSCs from conventional cultivation. * *p* < 0.05.

**Figure 4 medicina-60-01741-f004:**
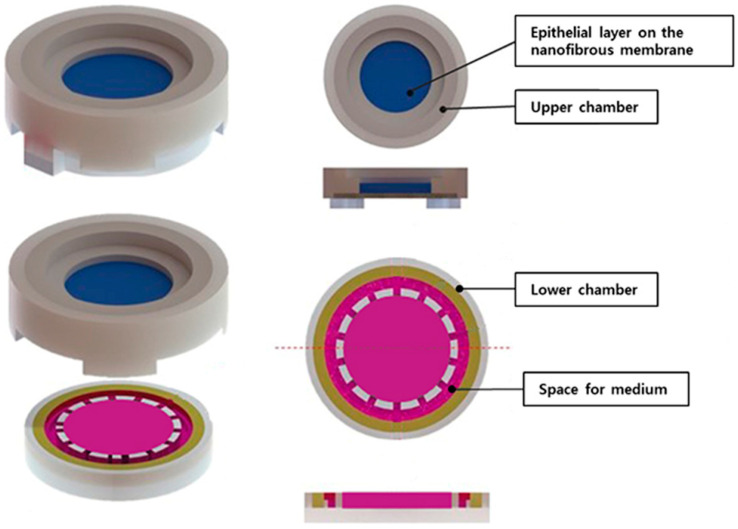
Schematic diagram of developed respiratory mucosa-on-a-chip model. The human respiratory mucosa-on-a-chip consisted of mucociliary epithelium and a stromal component that was fabricated in-house and designed to achieve an air–liquid interface.

**Figure 5 medicina-60-01741-f005:**
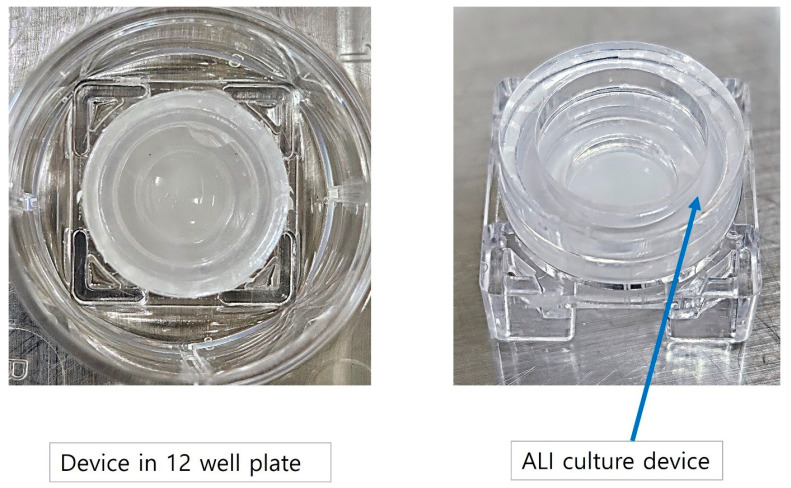
Respiratory mucosa-on-a-chip model in 12-well plate. The air–liquid interface culture device was placed in the 12-well plate (**left**) and three parts of the mucosa-on-a-chip were connected for respiratory differentiation (**right**).

**Figure 6 medicina-60-01741-f006:**
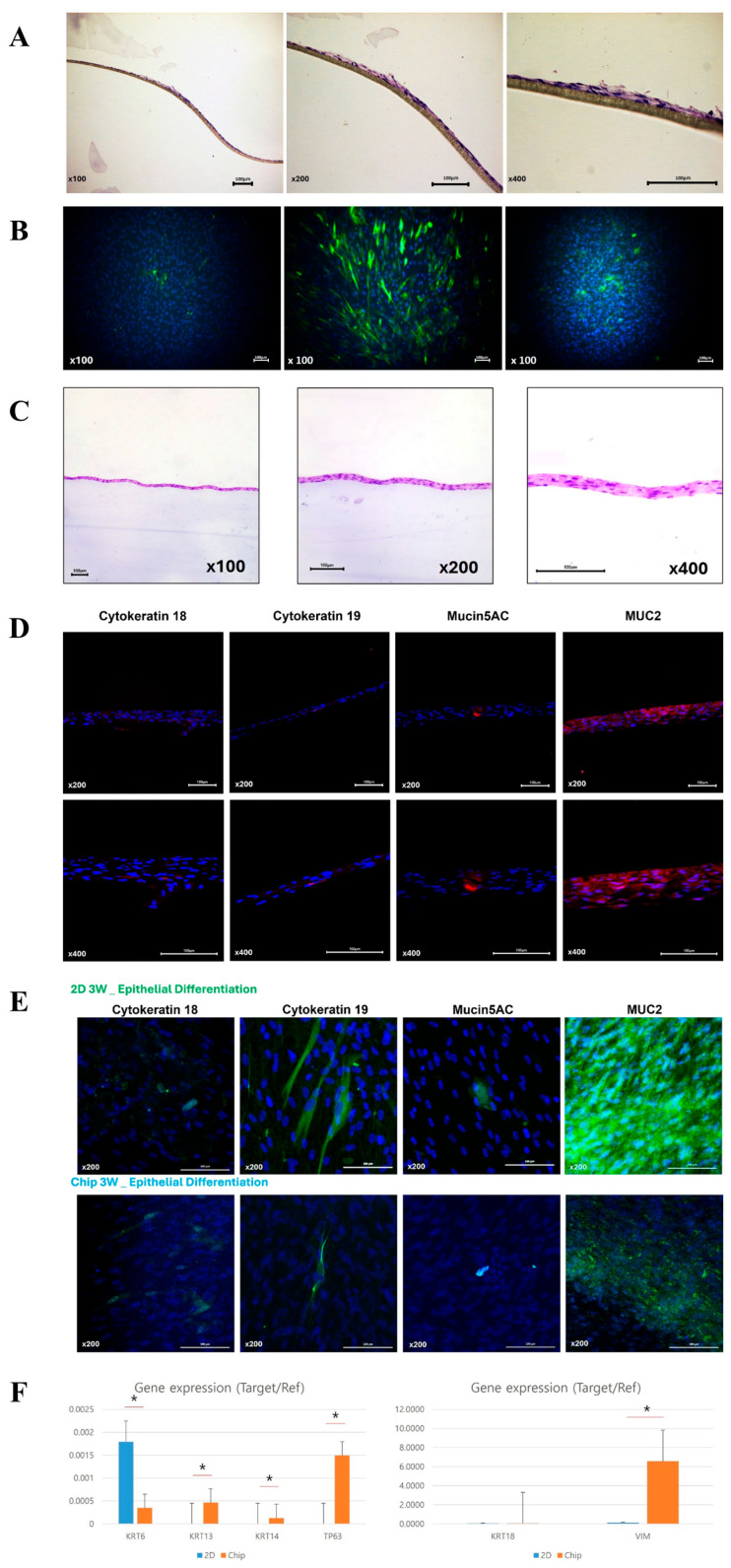
Comparison of the respiratory differentiation potential of hNTSCs from mucosa-on-a-chip and conventional culture groups. In the first preliminary tests, hNTSCs were cultivated in the mucosa-on-a-chip in growth media for 2 weeks (**A**), and these cells were cultivated under the respiratory differentiation conditions of a conventional 2D culture for 3 weeks. (**B**). In the mucosa-on-a-chip with respiratory differentiation, hNTSCs proliferated well with multi-layered growth and attachment to the membrane of the chip (**C**). In the immunofluorescence staining, hNTSCs were stained with cytokeratin 18, cytokeratin 19, MUC2, and mucin 5AC (**D**). hNTSCs in two groups were stained with cytokeratin 18, cytokeratin 19, MUC2, and mucin 5AC, respectively (**E**). In the quantitative analysis of gene expression by RT-PCR, there were significant differences in the differentiation capacity of hNTSCs derived from both conditions. Cytokeratin genes (KRT 13 and 14), transformation-related protein P63 (TP63), and vimentin of basal cells in the airway epithelium in the mucosa-on-a-chip group were significantly upregulated compared with those in the control group. In contrast, cytokeratin genes (KRT6) in the mucosa-on-a-chip group were significantly downregulated compared with those in the control group (**F**). * *p* < 0.05.

**Table 1 medicina-60-01741-t001:** Gene expression assays used for real-time polymerase chain reaction for respiratory differentiation.

Gene	Abbreviation	Reference Sequence	Assay Number
Keratin 6	KRT6	NM_173086	Hs00752476_s1
Keratin 13	KRT13	NM_002274	Hs01002802_m1
Keratin 14	KRT14	NM_000526	Hs00265033_m1
Keratin 18	KRT18	NM_000224	Hs02827483_g1
Transformation-related protein 63	TP63	NM_001114978	Hs00978340_m1
Vimentin	VIM	NM_003380	Hs00185584_m1
Glyceraldehyde-3-phosphate	GAPDH	NM_002046	Hs99999905_m1

## Data Availability

Information excluding patient-related information may be provided upon request to the corresponding authors.

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
