# Peer review of "The Development of a Human Respiratory Mucosa-on-a-Chip Using Human Turbinate-Derived Mesenchymal Stem Cells"

_medicina, 2024, doi:10.3390/medicina60111741_

Round 1
Reviewer 1 Report
Comments and Suggestions for Authors
The authors present a novel approach to create a respiratory mucosa-on-a-chip model using human nasal turbinate-derived stem cells (hNTSCs). The study offers a valuable contribution to the field of respiratory research, particularly in exploring the differentiation potential of hNTSCs in a more physiologically relevant microfluidic environment.
Here is my suggestion to the authors:
1. Figure 1B is not readable. Please consider put it in separate figure.
2. The sample size of five patients is relatively small and may not fully capture the variability in hNTSCs from different individuals. A larger cohort would strengthen the validity of the findings.
3. The study lacks a discussion on the potential limitations of the mucosa-on-a-chip model, such as the absence of immune cells or other physiological components that are present in native respiratory tissues.
4. The authors briefly mention the role of hNTSCs in regenerative medicine but do not explore the potential challenges in translating this model into clinical applications. Further discussion on the scalability and practical use of the model would be beneficial.
Author Response
Reviewer 1:
- Figure 1B is not readable. Please consider put it in separate figure.
◎ Reply:
We split Figure 1 into Figure 1 and Figure 2. You can see a larger figure in the original, not the PDF version.
- The sample size of five patients is relatively small and may not fully capture the variability in hNTSCs from different individuals. A larger cohort would strengthen the validity of the findings.
◎ Reply:
Most previous studies using microfluidics were conducted using a single donor (Lab Chip. 2024 Mar 12;24(6):1794-1807.; Front Bioeng Biotechnol. 2019 Jan 22:7:3; Lab Chip. 2018 May 1;18(9):1298-1309; Toxicol Res (Camb). 2018 Aug 11;7(6):1048-1060; Lab Chip. 2017 May 2;17(9):1578-1584; Biomed Microdevices. 2009 Oct;11(5):1081-9). Rather, it is very rare for microfluidics studies to use multiple donors for analysis. But from the perspective of the analysis itself, five donors may be insufficient. Therefore, we described the small sample size as a limitation of the study.
- The study lacks a discussion on the potential limitations of the mucosa-on-a-chip model, such as the absence of immune cells or other physiological components that are present in native respiratory tissues.
◎ Reply:
According to the reviewer’s comment, we listed the limitation of this study as below (Page 10 to 11):
Using a novel human respiratory mucosa-on-a-chip model, we were able to generate a model with higher cytokine activity functionally than conventional differentiation methods. This model is anticipated to be a valuable tool for assessing the respiratory mucosa's responses to external bacteria, viruses, and inhaled medications. How-ever, this model also has limitations. First, this chip was intended to mimic respiratory mucosa, but submucosa tissue was not implemented. It was constrained by the absence of other physiological components, such as immune cells, that are present in native tissues. In the actual human submucosa, small vessels exist, and through these, immune cells and inflammatory cells interact with each other. Additionally, glands such as Bowman's glands exist in the submucosa and secrete mucus, but these are absent in this model simulating the epithelium. However, there are still limitations in implementing such real human responses in microfluidics. Several studies have attempted to induce interactions by co-culture with immune cells or by creating microvascular structures in the substructure, but there are still limitations in producing responses similar to those in the human body. Second, this initial experiment was limited by an insufficient sample size. Despite these limitations, the model is anticipated to be valuable for evaluating the respiratory mucosa's response to external bacteria, viruses, and inhaled drugs.
- The authors briefly mention the role of hNTSCs in regenerative medicine but do not explore the potential challenges in translating this model into clinical applications. Further discussion on the scalability and practical use of the model would be beneficial.
◎ Reply:
We have added the following to the discussion section regarding the points raised by the reviewer: (Page 10):
The translation of the respiratory mucosa-on-a-chip model into clinical applications presents several significant challenges. First, biological complexity is a primary concern. While the chip can replicate certain physiological features of the respiratory mucosa, the human respiratory system involves a highly dynamic interplay of immune responses, microbial interactions, and variable environmental exposures. Replicating this complexity on a chip, particularly in a way that accounts for the heterogeneity of patient populations, remains difficult. Another challenge is maintaining long-term vi-ability and functionality of the cellular components on the chip, as in vitro models of-ten deteriorate over time, impacting their reliability for prolonged testing and therapeutic evaluation.
Another potential barrier is regulatory approval and validation. Before clinical use, any in vitro model must undergo rigorous testing to demonstrate its predictive accuracy in human conditions. This requires extensive comparative studies between the chip model and human trials, which can be time-consuming and costly. Furthermore, there are ethical and safety concerns regarding the use of these models to simulate human diseases, especially when considering immunocompromised or highly vulnerable patient groups.
Regarding the scalability and practical use of the respiratory mucosa-on-a-chip, there are technical and logistical challenges to be addressed. Manufacturing the chips at a scale necessary for widespread clinical application involves overcoming obstacles related to standardization and quality control. Each chip should consistently replicate the same biological conditions to ensure reproducibility across experiments and applications. Moreover, cost is a significant factor. Producing highly sophisticated and reliable models at an affordable price point for widespread clinical or pharmaceutical use is not trivial, particularly as the models become more complex to simulate specific patient conditions.
Lastly, the integration of this model into current clinical workflows poses practical concerns. Healthcare facilities would need to be equipped with the technical expertise and equipment to utilize these chips effectively, which may not be feasible in all clinical settings. Training personnel and integrating the chip model into routine diagnostic or therapeutic procedures will require time and resources, further complicating its transition to clinical use.
Reviewer 2 Report
Comments and Suggestions for Authors
In this manuscript authors have shown human respiratory mucosa-on-a-chip allows respiratory differentiation of hNTSCs and shows the difference of gene and cytokine expression. Although impressive, the authors are requested to address the following comments to make the manuscript technically sound.
Major comments:
1. Authors have not provided any diagrams or gross images on the microfluidic device / chip. It would be critical from the reader’s point of view to provide these details for a better understanding of the study.
2. Authors are requested to provide compositional details on electrospun nanofibrous membrane used in the middle layer of chip.
3. Figure 3B: it is unclear whether images are shown with different magnification or different staining markers. Please indicate clearly what each column represents.
4. As shown in figure 3E, it is unclear whether hNTSCs cultured on chip are better differentiated than 2D culture. Although the gene expression of these markers is slightly better in 3D group, the protein turn-out seems better in 2D culture. Authors are requested to address this issue and provide clear evidence showing one approach is better than the other.
5. Authors have claimed that the mucosa-on-chip provide better expansion of hNTSCs in abstract: “chip human respiratory mucosa-on-a-chip using human turbinate derived mesenchymal stem cells allows good expansion…” However, there is no data in the manuscript indicating mucosa-on-chip provides “better expansion” of cells. Authors are requested to provide relevant data to support their claims.
6. Sample size is missing in the statistical analysis description. Add data points in the bar graphs in all figures.
Minor comments:
1. Fig 1 caption: (hNTSCs) are described as human neural stem cells instead of human nasal inferior tur-binate-derived stem cells.
2. Scale bar is missing in several figures. Authors are requested to add scale bar instead of simply stating 100X / 400X magnification on the figures for better understanding of scale of these samples.
3. Figure 3F: Show gene expression as Log2 fold change for a clearer presentation of data.
Author Response
Reviewer 2:
Major comments:
- Authors have not provided any diagrams or gross images on the microfluidic device / chip. It would be critical from the reader’s point of view to provide these details for a better understanding of the study.
◎ Reply:
We added a gross image of the device and its description (Figure 4).
- Authors are requested to provide compositional details on electrospun nanofibrous membrane used in the middle layer of chip.
◎ Reply:
We specifically describe the membrane fabrication method as follows (Page 3):
The electrospun nanofibrous membrane was positioned between the upper and lower chambers. The membrane was fabricated in the following manner: polycaprolactone was dissolved in a 9:1 chloroform-dimethylformamide solution and subsequently elec-trospun into nanofibers with diameters ranging from 300 to 500 nm, resulting in the formation of a nanofibrous membrane with a thickness of 200 µm. The electrospinning process was conducted at room temperature, utilizing a flow rate of 2 ml/h and an applied voltage of 18 kV. The inner diameter of the needle was 0.4 mm, and the distance between the needle tip and the collecting plate was maintained at 15 cm.
- Figure 3B: it is unclear whether images are shown with different magnification or different staining markers. Please indicate clearly what each column represents.
◎ Reply:
All images are at the same scale of x100. Two images were missing the scale indicator, so we added it (Figure 5B).
- As shown in figure 3E, it is unclear whether hNTSCs cultured on chip are better differentiated than 2D culture. Although the gene expression of these markers is slightly better in 3D group, the protein turn-out seems better in 2D culture. Authors are requested to address this issue and provide clear evidence showing one approach is better than the other.
◎ Reply:
Whether or not certain markers are expressed might be an issue, but a more intense fluorescent staining does not always mean a better model. We added that while the fluorescence staining results may appear similar between 2D culture and chip models, the chip model was able to elicit more diverse gene expression trends in terms of gene expression trends (Page 9).
- Authors have claimed that the mucosa-on-chip provide better expansion of hNTSCs in abstract: “chip human respiratory mucosa-on-a-chip using human turbinate derived mesenchymal stem cells allows good expansion…” However, there is no data in the manuscript indicating mucosa-on-chip provides “better expansion” of cells. Authors are requested to provide relevant data to support their claims.
◎ Reply:
The phrase "good" has been removed as it may be misleading.
- Sample size is missing in the statistical analysis description. Add data points in the bar graphs in all figures.
◎ Reply:
We collected samples from five patients, which are described in the abstract and methods sections.
Minor comments:
- Fig 1 caption: (hNTSCs) are described as human neural stem cells instead of human nasal inferior tur-binate-derived stem cells.
◎ Reply:
We corrected "neural" to "nasal".
- Scale bar is missing in several figures. Authors are requested to add scale bar instead of simply stating 100X / 400X magnification on the figures for better understanding of scale of these samples.
◎ Reply:
Figure 5A, 5B images cannot be inserted arbitrarily because the scale bar was missing when the images were taken. The images with missing magnification were inserted by adding the magnification.
- Figure 3F: Show gene expression as Log2 fold change for a clearer presentation of data.
◎ Reply:
Since the differences in values are significant for gene expression-related content, it is sufficient to present them in intuitive figures.
Round 2
Reviewer 1 Report
Comments and Suggestions for Authors
The author has answered all my questions. The manuscript look good and is ready to publish.
Author Response
We thank the reviewer for his thoughtful review of the manuscript.
Reviewer 2 Report
Comments and Suggestions for Authors
Authors have not satisfactorily addressed major comments . Especially points #4, 5, 6 and minor #3
. The gross image of 3D chip does not provide efficient design details. Authors should provided a detailed diagram with all components clearly pointed out.
Author Response
Reviewer 2:
Authors have not satisfactorily addressed major comments. Especially points #4, 5, 6 and minor #3.
- As shown in figure 3E, it is unclear whether hNTSCs cultured on chip are better differentiated than 2D culture. Although the gene expression of these markers is slightly better in 3D group, the protein turn-out seems better in 2D culture. Authors are requested to address this issue and provide clear evidence showing one approach is better than the other.
◎ Reply:
As the reviewer commented, 2D showed higher results in staining and 3D showed higher results in gene expression. This shows that the form of differentiation proceeds differently depending on the differentiation environment between the two methods, but does not mean that one method is superior. It would be necessary to elucidate these differences in protein expression and genetic expression through additional studies in the future.
We have added the above comment to the text (Page 9).
- Authors have claimed that the mucosa-on-chip provide better expansion of hNTSCs in abstract: “chip human respiratory mucosa-on-a-chip using human turbinate derived mesenchymal stem cells allows good expansion…” However, there is no data in the manuscript indicating mucosa-on-chip provides “better expansion” of cells. Authors are requested to provide relevant data to support their claims.
◎ Reply:
Our intention was not to mean proliferation as in “better expansion” This meant that it could proliferate and differentiate without problems, and the results (immunostaining and gene expression) could be confirmed. Therefore, to prevent confusion, the “expansion” phrase was also deleted (Page 1).
- Sample size is missing in the statistical analysis description. Add data points in the bar graphs in all figures.
◎ Reply:
We have added the following to the statistical analysis section (Page 5):
We conducted a study with cells donated from 5 participants, and the sample size for all results was 5.
We added scale bars to Figures 1, 5A and 5B that were missing scale bars.
Minor comments:
- Figure 3F: Show gene expression as Log2 fold change for a clearer presentation of data.
◎ Reply:
The differences in values related to gene expression are significant, and calculations can be difficult, especially since the denominator is close to 0.
The gross image of 3D chip does not provide efficient design details. Authors should provided a detailed diagram with all components clearly pointed out.
◎ Reply:
We have added the schematic diagram of developed respiratory mucosa-on-a-chip model as Figure 4.